# Variation in Certain Soil Properties Based on Land Use Type, and Elevation in Arhavi Sub-Basin, Artvin, Turkiye

Ahmet Duman [1,*], Cengizhan Yildirim [2], Mustafa Tufekcioglu [3], Aydın Tufekcioglu [3] and Caner Satiral [4]

1  Department of Forestry, Artvin Vocational School, Artvin Coruh University, Artvin 08100, Türkiye
2  Department of Forest Engineering, Institute of Graduate Education, Artvin Coruh University, Artvin 08100, Türkiye; cengizhanyildirim@ogrenci.artvin.edu.tr
3  Department of Forest Engineering, Faculty of Forestry, Artvin Coruh University, Artvin 08100, Türkiye; mtufekcioglu61@artvin.edu.tr (M.T.); atufekci@artvin.edu.tr (A.T.)
4  Forestry Application and Research Center, Artvin Coruh University, Artvin 08100, Türkiye; csatiral@artvin.edu.tr
*  Correspondence: ahmetbozyazi33@artvin.edu.tr; Tel.: +90-0533-301-7295

**Abstract:** Knowledge of soil properties such as texture, bulk density, organic matter, soil reaction, soil electrical conductivity, and soil erodibility factor is fundamental to the sustainable management of soil resources. This study aimed to determine the changes in certain soil properties including texture, bulk density, organic matter, pH, electrical conductivity, and soil erodibility factor with different land uses, elevation zones and soil depths in the Arhavi Sub-basin, Artvin, Turkiye. For these purposes, a total of 618 soil samples (309 disturbed and 309 undisturbed) were taken from 155 sampling points located in areas with different land uses, including tea (33 sampling points), hazelnut (33 sampling points), forest (67 sampling points), and grassland (22 sampling points). The results of the statistical analysis revealed that the soils at depths of 0–15 cm and 15–30 cm showed significant differences according to the land use in terms of sand, clay, silt, bulk density, organic matter, and pH. There were significant positive correlations between elevation and soil bulk density ($r = 0.495$) at a soil depth of 0–15 cm in the tea areas. In the grassland areas, there were significant positive correlations between elevation and silt, bulk density, and the soil erodibility factor ($r = 0.485$, $r = 0.794$, and $r = 0.442$, respectively) at depths of 0–15 cm, and significant positive correlations between elevation and both silt and bulk density ($r = 0.468$ and $r = 0.691$, respectively) at depths of 15–30 cm. Similarly, there were significant positive correlations between elevation and both sand and organic matter at soil depths of 0–15 cm and 15–30 cm (sand: $r = 0.351$ and $r = 0.638$, respectively; organic matter: $r = 0.277$ and $r = 0.587$, respectively). On the other hand, significant negative correlations were found between elevation and silt, bulk density, pH, and the soil erodibility factor at depths of 0–15 cm and 15–30 cm (silt: $r = -0.400$ and $r = -0.461$, respectively; bulk density: $r = -0.593$ and $r = -0.545$, respectively; pH: $r = -0.354$ and $r = -0.309$, respectively; soil erodibility factor: $r = -0.443$ and $r = -520$, respectively). Soil acidity was found to be the most important problem threatening soil fertility in all land uses. The use of soil acidity-increasing fertilizers, such as ammonium sulfate, in tea gardens in the region should be eliminated to protect the fertility of soils in the future. The knowledge that this study provides might help farmers and foresters in the region in the proper management and fertilization of their lands. Moreover, this study will provide data to future studies related to soil acidification, soil erosion, and land use that are planned for the Arhavi Sub-basin.

**Keywords:** Arhavi; soil acidity; elevation zones; land-use type; soil depth; black sea region

## 1. Introduction

The sustainable use of natural resources such as forests, grasslands, and croplands is the key to keeping these ecosystems healthy [1,2]. However, growing populations increase the pressure on natural resources day by day [1]. This causes the unconscious, excessive, or

incorrect use of natural resources, resulting in reduced productivity or jeopardized sustainability. In Turkiye, especially in regions where agriculture, forestry, animal husbandry, and grassland practices are widely utilized, grassland and forest areas are degraded as a result of early grazing and overgrazing [1]. On the other hand, agricultural areas are degraded as a result of increased surface erosion through improper land use and agricultural practices [1,2]. The degree of harm caused to ecosystems varies with the type of land use, soil type, and elevation [1,2].

Soil properties vary also with the type of land use, elevation, and soil depth. There are many national and international scientific studies showing that certain soil properties show statistically significant differences with different types of land use [1–20]. Turudu [8] reported that the clay content in agricultural soils was higher than in meadow and grassland soils. Similarly, Yener et al. [9] stated that the clay content of soils varied according to the soil depth and was lower in alder forest soils than in tea soils.

Changes in elevation affect soil organic matter by controlling plant species and biomass production in natural and cultural areas, soil-water balance, soil erosion, and geological deposition processes [21]. Tekeş and Cürebal [22] reported that land use varies with elevation, with agriculture and settlement more common at lower elevations, while forest and grassland areas are more common at higher elevations. There have been few studies performed in the Eastern Black Sea region on changes in soil properties with land use, elevation, and soil depth [5,8,9,23–28]. Thus, there are still gaps in the research in terms of how soil properties change in tea, hazelnut, forest, and grassland sites with elevation in this region of Turkiye. The knowledge that the present study will provide may help foresters and farmers in the region, especially, in the proper management and fertilization of their land.

The objective of this study was to assess changes to some soil physiochemical properties (texture, bulk density (BD), organic matter (OM), electrical conductivity (EC), soil pH, and soil erodibility factor (K)) with different types of land use (tea, hazelnut, grassland, and forest areas), elevation zones, and soil depths in the Arhavi Sub-basin.

We hypothesized that the physiochemical properties of soils in the Arhavi Sub-basin would vary based on the type of land use, elevation, and soil depth, with the forest and grassland areas having better soil physiochemical properties than tea and hazelnut areas.

## 2. Materials and Methods

### 2.1. Study Area and Sampling Locations

The study area is located at the Arhavi Sub-basin with a surface area of 300 km$^2$, between 41°07′00″–41°21′30″ north and 41°15′00″–41°30′00″ east within the Eastern Black Sea region in northeastern Turkiye (Figure 1).

The elevation of the Arhavi Sub-basin ranges between 20 and 3343 m, with an average elevation of 1672 m above sea level. The study area is thus located in high mountainous land. The average total annual precipitation of the study area is 2438 mm and the average annual temperature is 11.8 °C. The area has a typical, very humid Black Sea climate with abundant rains in every season [29].

The geology of the study area consists of volcanic facies from the Upper Cretaceous and Eocene periods, granite, granodiorite and quartz diorites from the Tertiary period, new formations from the Holocene period, and glaciers and moraines from the Pleistocene period [30]. The soils at the study site are mainly well-drained ultisols at low elevations and mainly well-drained podzols at high elevations, and they have textures ranging from sandy loam to clay loam.

In terms of land use, 17,450 ha (58.2%) of the study area is forested [31], 5124 ha (17.1%) are used for tea garden [32], 700.8 ha (2.3%) are for hazelnut [33], 3668 ha (12.2%) are grassland, and the remaining 3058 ha (10.2%) are other areas (garden, agricultural residences, roads, streams, rocky areas, various facilities, etc.).

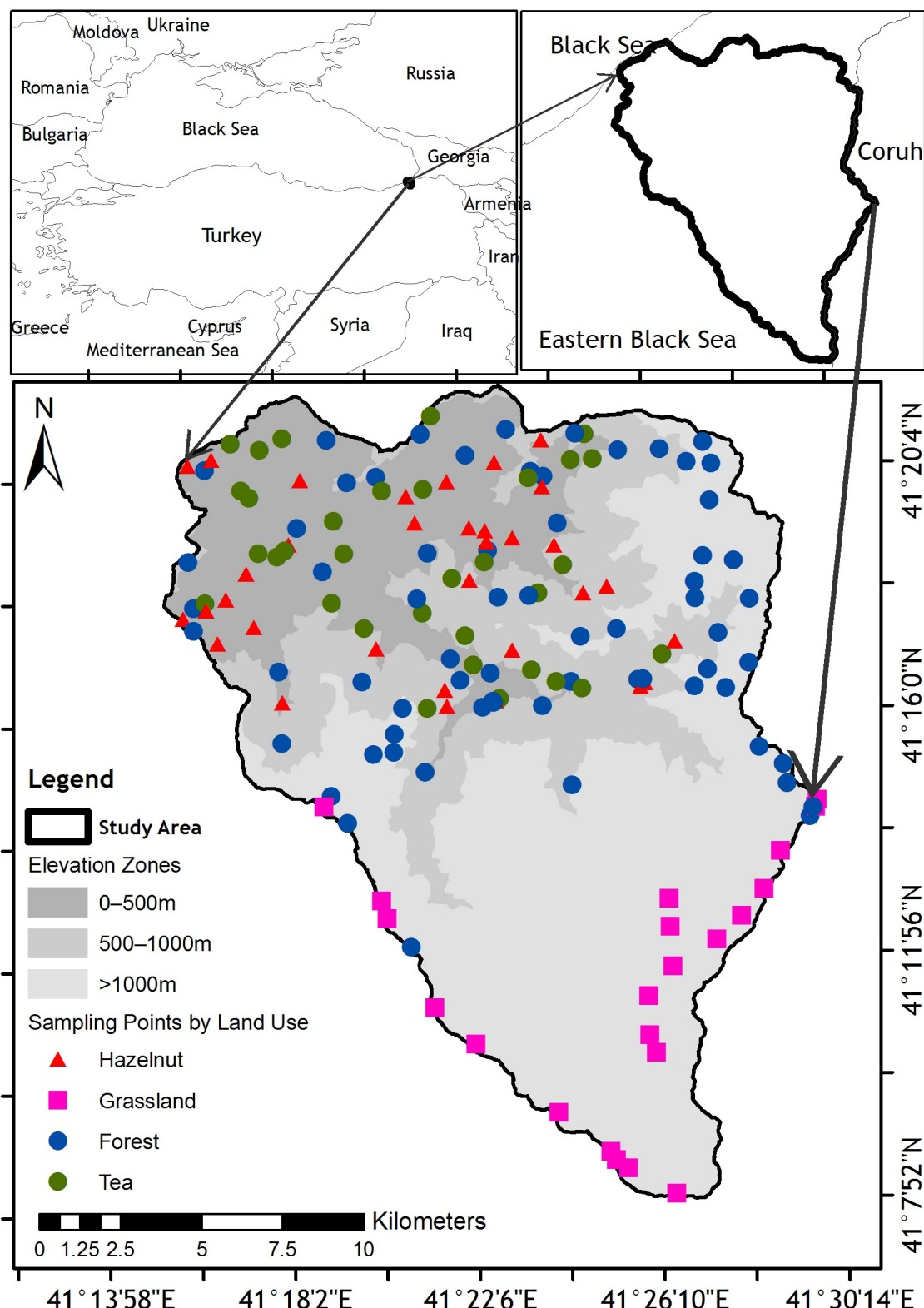

**Figure 1.** Study Area and Soil Sampling Points.

Some photographs of different land uses in the study area are given in Figure 2.

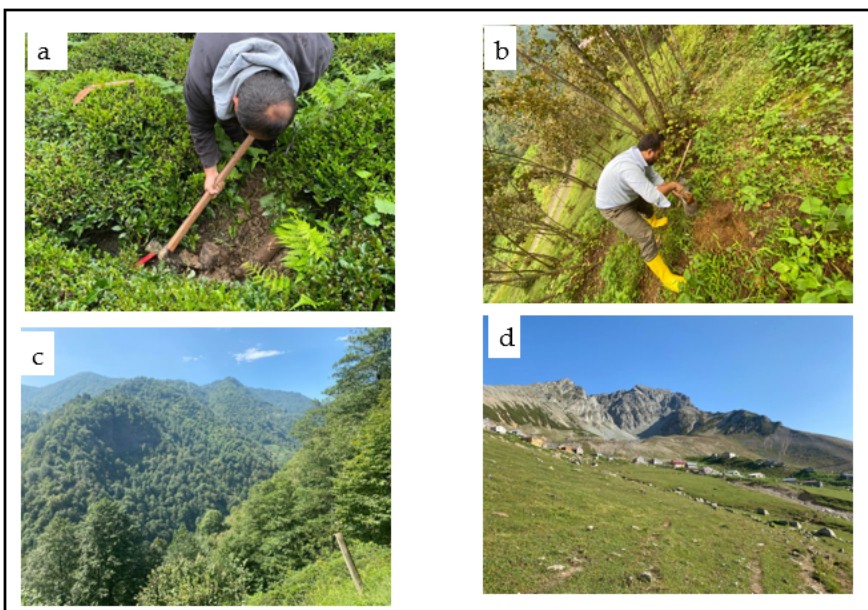

**Figure 2.** Some photographs of different land uses in the study area: (**a**) tea, (**b**) hazelnut, (**c**) forest, and (**d**) grassland.

### 2.2. Sampling and Analysis Methods

Soil sampling points were determined according to a stratified simple random sampling method [34]. Accordingly, in 2020, a total of 618 soil samples (309 disturbed and 309 undisturbed) were taken from 155 sampling points located in areas of different land uses, including tea (33 sampling points), hazelnut (33 sampling points), forest (67 sampling points), and grassland (22 sampling points). The disturbed soil samples were air-dried at room temperature, ground in a mortar, sieved through a 2 mm sieve, and prepared for analysis. The textures of soil samples were determined according to Bouyoucos' hydrometer method [35]; pH and electrical conductivity were determined according to the glass electrode method in a 1:2.5 soil–water mixture [36]; organic matter according to the modified Walkley–Black wet combustion method [37], and soil erodibility (K factor in RUSLE) according to the indirect estimation method using texture and organic matter content as independent variables with the help of an equation developed by Sharpley and Willams [38], found in Equation (1).

$$
\begin{aligned}
\mathrm{K} = &\left(0.2 + 0.3\ \exp\left[-0.256\ \mathrm{SAN}\ \left(1 - \tfrac{\mathrm{SIL}}{100}\right)\right]\right) \times \left[\tfrac{\mathrm{SIL}}{\mathrm{CLA+SIL}}\right]^{0.3} \times \left[1.0 - \tfrac{0.25\ \mathrm{C}}{\mathrm{C} + \exp\left[(3.72 - 2.95\ \mathrm{C})\right]}\right] \times \\
&\left[1.0 - \tfrac{0.7\ \left(1 - \tfrac{\mathrm{SAN}}{100}\right)}{\left(1 - \tfrac{\mathrm{SAN}}{100}\right) + \exp\left[-5.51 + 22.9\ \left(1 - \tfrac{\mathrm{SAN}}{100}\right)\right]}\right] \times 0.1317,
\end{aligned}
\tag{1}
$$

SAN: Sand (%), SIL: Silt (%), CLA: Clay (%), C: Organic Matter (%).

### 2.3. Data Analysis

The SPSS 19.0 package program was used for data processing and evaluation [39]. A one-way ANOVA was used to determine whether soil properties differed according to land uses and elevation zones. Pairwise comparisons of the groups were made according to the results of Duncan's *t*-test when the variances were equal and Tamhane's T2 test when the variances were not equal. A comparison of soil properties based on depth level was made according to the results of an independent samples *t*-test.

## 3. Results and Discussion

### 3.1. Soil Texture

Figure 3 shows the percentage distributions of soil texture classes according to land use, elevation zone, and depth levels. According to these results, there were many soil

texture classes found in the study area, which ranged from sandy loam to clay loam textures, with sandy loam, sandy clay loam, and loamy sand being the most common. Similarly, Yüksek et al. [24] reported that the soil texture of a tea plantation site was sandy clay loam, and the soil texture of an alder coppice site was loamy sand in a study carried out in the region (Pazar-Rize). On the other hand, in a study carried out by Yener et al. [9], it was reported that soils with loam, loamy clay, and sandy loam textures were common in the same region.

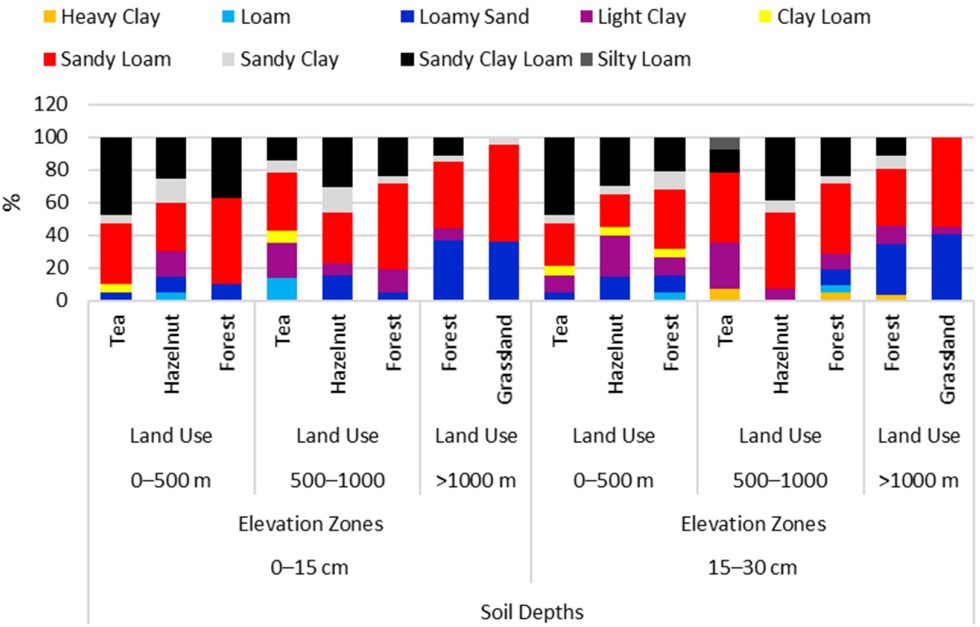

**Figure 3.** Distribution of soil texture classes by land use, elevation, and soil depth in the study area.

3.1.1. Sand (%)

Variations in the mean sand content of the soils delineated by land use, elevation zone, and soil depth levels are given in Figure 4. The sand content of the soils at 0–15 cm depths differed significantly according to land use ($p < 0.05$). Pairwise comparisons demonstrated that the sand content at 0–15 cm soil depths in the 500–1000 m elevation zone was significantly lower ($p < 0.05$) in tea areas (51.49%) than in hazelnut (61.70%) and forest (61.47%) areas. The reason behind this could be that tea areas have less surface erosion compared to forest and hazelnut areas because terraces in tea areas decrease surface erosion. Similar results are found in studies performed in the region by Yüksek et al. [24] and Karagül [25] and in other parts of the world by Molla et al. [10], Asmare et al. [11], and Negasa et al. [19]. Karagül [25] suggested that the differences in sand content with different land uses could be explained by the fact that changes in topography and elevation in the basin affect weathering, leaching, and profile development.

The sand content of soils at 0–15 cm depths differed significantly according to elevation zone ($p < 0.05$). Pairwise comparisons revealed that the clay content of soils at 0–15 cm depths was significantly higher in forest areas located at >1000 m (70.03%) elevation than those located at 0–500 m (63.63%) and 500–1000 m (61.47%) ($p < 0.05$). This might be a result of precipitation increasing with elevation. The results of the correlation analysis revealed significant positive correlations between elevation and sand content at 0–15 cm and 15–30 cm soil depths in forest areas ($p < 0.05$) (Table 1). In contrast to our findings, Tellen et al. [40] reported no significant relationship between sand content and elevation in forest areas.

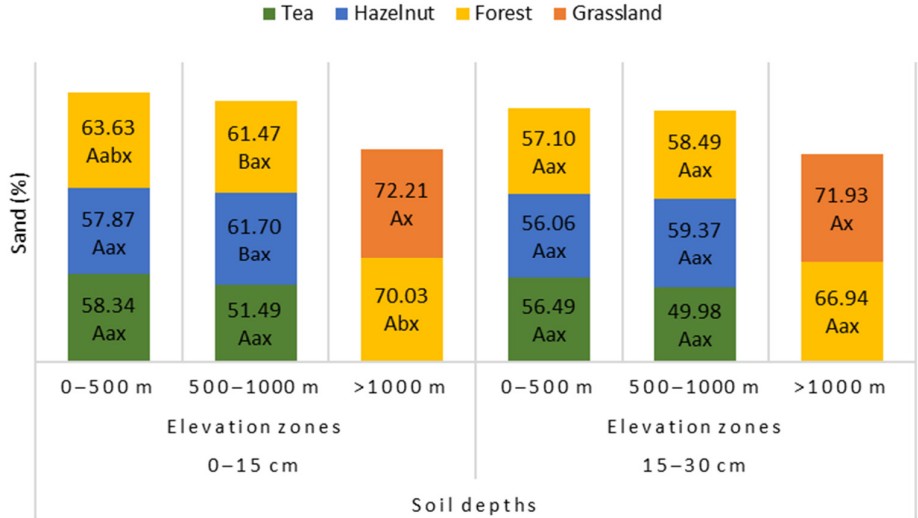

**Figure 4.** Changes in soil sand (%) content with land use, elevation, and soil depth. Letters A and B indicate significant differences among the land uses at each depth. Letters a and b indicate variation among the elevation zones at each soil depth. Letter x indicate variation among the soil depths in each land use and elevation zone.

**Table 1.** Correlation coefficients (r) between elevations and some soil properties for different land uses at two soil depths (0–15 cm and 15–30 cm).

| Land Use Type | Tea | | Hazelnut | | Grassland | | Forest | |
|---|---|---|---|---|---|---|---|---|
| Soil Depth/ Soil Properties | 0–15 cm | 15–30 cm | 0–15 cm | 15–30 cm | 0–15 cm | 15–30 cm | 0–15 cm | 15–30 cm |
| Sand (%) | −0.31 | −0.2 | 0.219 | 0.128 | −0.404 | −0.294 | 0.351 ** | 0.277 * |
| Clay (%) | 0.198 | 0.139 | −0.136 | −0.068 | 0.125 | 0.022 | −0.229 | −0.006 |
| Silt (%) | 0.243 | 0.163 | −0.271 | −0.198 | 0.485 * | 0.468 * | −0.400 ** | −0.461 ** |
| BD (g/cm$^3$) | 0.493 ** | 0.094 | −0.012 | −0.003 | 0.794 ** | 0.691 ** | −0.593 ** | −0.545 ** |
| OM (%) | −0.007 | 0.104 | 0.275 | 0.229 | −0.292 | −0.12 | 0.638 ** | 0.587 ** |
| pH | 0.283 | 0.239 | 0.343 | 0.27 | 0.14 | 0.102 | −0.354 ** | −0.309 * |
| EC (µS/cm) | 0.08 | −0.286 | 0.221 | 0.291 | 0.235 | 0.371 | 0.148 | 0.154 |
| K (ton ha h ha$^{-1}$ MJ$^{-1}$ mm$^{-1}$) | 0.012 | 0.155 | −0.049 | −0.05 | 0.442 * | 0.4 | −0.443 ** | −0.520 ** |

** Correlation is significant at the 0.01 level (2-tailed). * Correlation is significant at the 0.05 level (2-tailed). BD: bulk density, OM: organic matter, EC: electrical conductivity, K: soil erodibility factor.

Sand content in the tea fields decreased at both depths with the increase in elevation. This result is in contrast with Yuksek et al.'s [26] findings in a neighboring city, Rize. They reported that the sand content of the soils increased in the tea areas with the increase in elevation. This difference might be due to differences in the soil texture, parent material, and the type of rainfall. While higher elevations experience precipitation mainly as snow in the winter time, lower elevations experience it as rain throughout the winter in these areas. The slow melting of snow may cause less leaching in the soil profile compared to rainfall.

### 3.1.2. Clay (%)

Variations in the mean clay content of soils with land use, elevation zone, and soil depth levels are given in Figure 5. The clay content of the soils at 15–30 cm depths varied significantly with land use ($p < 0.05$). It was found to be significantly lower in grassland areas (7.96%) than in forest areas (14.93%) in the >1000 m elevation zone ($p < 0.05$). Similar results have been found by Tufekcioglu and Kucuk [41] and Mulugeta et al. [20]. Similar

studies performed in the region include Yuksek et al. [24], which reported significantly lower clay content in forest areas than in tea areas; Turudu [8], which stated that the clay content of agricultural soils was higher than that of meadow and grassland soils; and Yener et al. [9], which reported that the clay content of soils varied according to soil depth and was lower in forest soils than in tea soils.

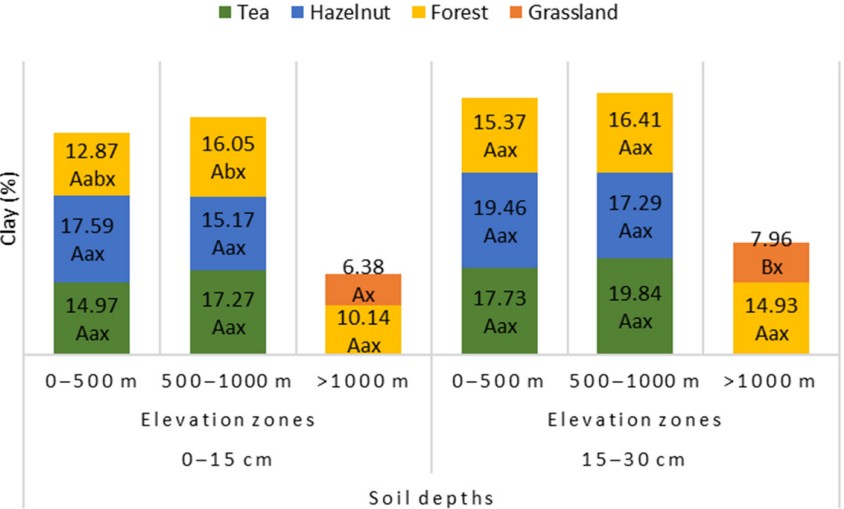

**Figure 5.** Changes in soil clay (%) content with land use, elevation, and soil depth. Letters A and B indicate significant differences among the land uses at each depth. Letters a and b indicate variation among the elevation zones at each soil depth. Letter x indicate variation among the soil depths in each land use and elevation zone.

The clay content of soils at 0–15 cm depths varied significantly with elevation zones ($p < 0.05$). Pairwise comparisons revealed that clay content of soils at 0–15 cm depths was significantly lower in forest areas located at the >1000 m (6.38%) elevation zone than those located at 0–500 m (12.87%) and 500–1000 m (16.05%) zones ($p < 0.05$). This could be the result of an increased leaching of clay particles in the soil profile as a result of the increase in precipitation with elevation. Similar results have been found by Tellen et al. [40].

The clay content in tea fields increased with the increase in elevation at both soil depths. In contrast to our results, a study conducted by Yuksek et al. [26] in the same region determined that the clay content of soils decreased with increasing elevation in tea fields. This difference might be due to the differences in the parent material, land use history, and soil texture properties.

3.1.3. Silt (%)

Variations in the average silt (%) content of the soils with land use, elevation zone, and soil depth levels are given in Figure 6. The silt content of the soils differed significantly with land use ($p < 0.05$). Similar results have been found by Mulugeta et al. [20]. Pairwise comparisons revealed that the silt content was significantly ($p < 0.05$) higher at 0–15 cm and 15–30 cm soil depths in tea areas (31.24% and 30.18%, respectively) than in hazelnut (23.13% and 23.33%, respectively) and forest (22.48% and 25.09%, respectively) areas in the 500–1000 m elevation zone.

The silt content of tea fields increased with elevation at both depth levels. This difference might be due to differences in the parent material, land use history, and soil texture properties. There was a significant positive correlation between silt content and elevation in grassland areas ($p < 0.05$). On the other hand, there was a significant ($p < 0.05$) negative correlation between silt and elevation in forest areas ($p < 0.01$) (Table 1). In addition, the silt content of the soils at 15–30 cm depths was significantly lower in forest areas located in the >1000 m elevation zone (18.14%) than those located in the 0–500 m (27.53%) and 500–1000 m (25.09%) zones ($p < 0.05$).

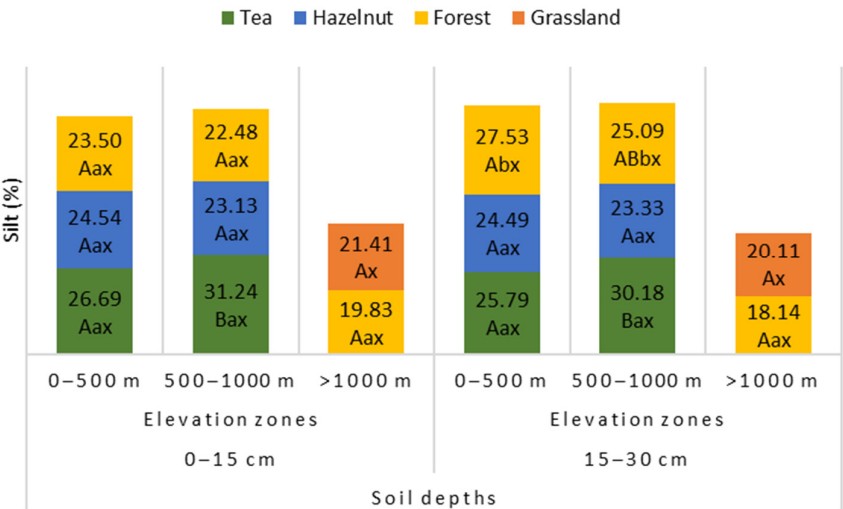

**Figure 6.** Changes in soil silt (%) content with land use, elevation, and soil depth. Letters A and B indicate significant differences among the land uses at each depth. Letters a and b indicate variation among the elevation zones at each soil depth. Letter x indicate variation among the soil depths in each land use and elevation zone.

### 3.2. Soil Bulk Density (g/cm³)

Variations in the average bulk density of soils with land use, elevation zone, and soil depth levels are given in Figure 7. The bulk density of the soils at 0–15 cm and 15–30 cm depths showed significant differences with land use ($p < 0.05$). According to the pairwise comparisons, the bulk density at 0–15 cm depths in the 0–500 m elevation zone was found to be significantly lower in tea areas (0.80 g/cm³) than in hazelnut areas (0.98 g/cm³) ($p < 0.05$), while the bulk density at 0–15 cm and 15–30 cm depths in the >1000 m elevation zone was found to be significantly lower in forest areas (0.52 g/cm³ and 0.71 g/cm³, respectively) than in grassland areas (0.77 g/cm³ and 0.86 g/cm³, respectively) ($p < 0.05$). The lower bulk density in forest and grassland areas compared to tea and hazelnut areas might be due to a higher organic matter content in the soils of these areas. The organic matter content of soil decreases with tillage and surface erosion in agricultural areas. The fact that the bulk density values of soil in grassland areas are higher than those in forest areas can be explained as a result of soil compaction due to intensive animal grazing in grassland areas. Shamsher et al. [42] found that the bulk density of soil differed significantly with land use. They reported that bulk density in forest areas was significantly lower than in grassland and agricultural areas at all depth levels, and that bulk density decreased with increasing elevation and increased with increasing soil depth. Kebebew et al. [43] reported that the bulk density of soil was significantly affected by type of land use. Tellen et al. [40] determined that bulk density differed significantly with land use.

The bulk density of soils at 0–15 cm and 15–30 cm depths showed significant differences with elevation zone and soil depth ($p < 0.05$). The bulk density of soils at 0–15 cm and 15–30 cm depths was found to be significantly lower in forest areas in the >1000 m elevation zone (0.52 g/cm³ and 0.71 g/cm³, respectively), than in forest areas at elevation zones of 0–500 m (0.83 g/cm³ and 1.05 g/cm³, respectively) and 500–1000 m (0.84 g/cm³ and 1.01 g/cm³, respectively) ($p < 0.05$). Similarly, Yüksek et al. [27] stated that the bulk density of soils in forest areas first decreased and then increased with an increase in elevation, while the bulk density of soils increased with an increase in soil depth. In addition, Yuksek et al. [26] determined that bulk density values increased in tea fields with an increase in elevation. Similar results have been reported by Tellen et al. [40] and Shamsher et al. [42].

The results of the correlation analysis showed that there were significant positive correlations between elevation and bulk density in soils at 0–15 cm depths in tea areas and 0–15 cm and 15–30 cm depths in grassland areas, and significant negative correlations

between elevation and bulk density in soils at 0–15 cm and 15–30 cm depths in forest areas ($p < 0.05$) (Table 1).

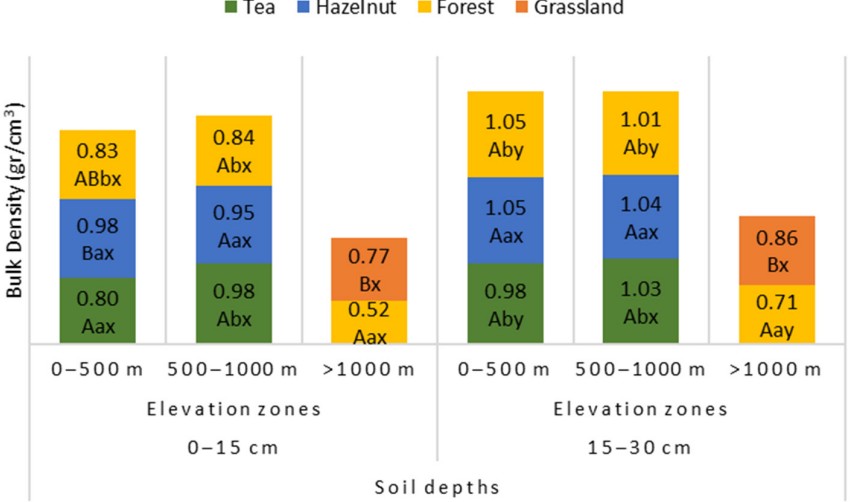

**Figure 7.** Changes in soil bulk density with land use, elevation, and soil depth. Letters A and B indicate significant differences among the land uses at each depth. Letters a and b indicate variation among the elevation zones at each soil depth. Letters x and y indicate variation among the soil depths in each land use and elevation zone.

### 3.3. Soil Organic Matter (%)

Variations in the average organic matter of soils with land use, elevation zone, and soil depth levels are given in Figure 8. The organic matter of soils at 0–15 cm and 15–30 cm depths showed significant differences with land use ($p < 0.05$). Further, the organic matter content of soils at 0–15 cm depths for all land uses in all elevation zones was found to be significantly higher than at 15–30 cm depths ($p < 0.05$). Shamsher et al. [42] determined that the organic matter content of soil differed significantly according to land use, that organic matter content in forest areas was significantly higher than in grassland and agricultural areas at all depth levels, and that organic matter content increased with an increasing elevation and decreased with an increasing soil depth. Similarly, Tellen et al. [40] and Khaldoon et al. [44] determined that the organic carbon of soil differed significantly according to the land use. Tüfekçioğlu [28] reported that the organic matter content of soils at the 0–30 cm depth level was greater in forest areas than in grassland areas.

The organic matter of soil differed significantly with elevation zone and soil depth ($p < 0.05$). The results of the correlation analysis revealed significant positive correlations between elevation and organic matter content in forest areas ($p < 0.01$) (Table 1). Further, the organic matter content of soils at 0–15 cm depths for all land uses in all elevation zones was found to be significantly higher than the organic matter content of soils at 15–30 cm depths ($p < 0.05$). Shamsher et al. [42] determined that the organic matter content of soil differed significantly according to elevation and soil depth. He reported that the organic matter content of soil in forest areas was significantly higher than in grassland and agricultural areas at all depth levels and increased with elevation and decreased with soil depth. Similarly, Yuksek et al. [26] found that the organic matter content of soils in tea gardens increased with elevation and differed significantly with soil depth ($p < 0.05$).

### 3.4. Soil pH

Variations in the average pH values of soils with land use, elevation zone, and soil depth levels are given in Figure 9. The pH values of soils at 0–15 cm and 15–30 cm depths showed significant differences with land use ($p < 0.05$). This result was consistent with the results of many other studies, indicating that the pH of soils differs significantly according to types of land use [5,42,44–46]. Pairwise comparisons revealed that pH values were

significantly lower at 0–15 cm soil depths in tea areas (3.35) than in hazelnut (4.00) and forest (4.09) areas in 0–500 m elevation zones ($p < 0.05$). Similar differences were also found at 15–30 cm soil depths in tea areas (3.51) that had a low pH value compared to hazelnut (4.10) and forest areas (4.20) located in the 0–500 m elevation zone. The pH values at 0–15 cm soil depths were significantly lower in tea areas located in the 500–1000 m elevation zones (3.77) than in hazelnut (4.23) areas, as well ($p < 0.05$). On the other hand, the pH values of soils at 0–15 cm and 15–30 cm soil depths were significantly lower in forest areas (3.67 and 3.86, respectively) than in grassland areas (4.08 and 4.18, respectively) in the >1000 m elevation zone ($p < 0.05$).

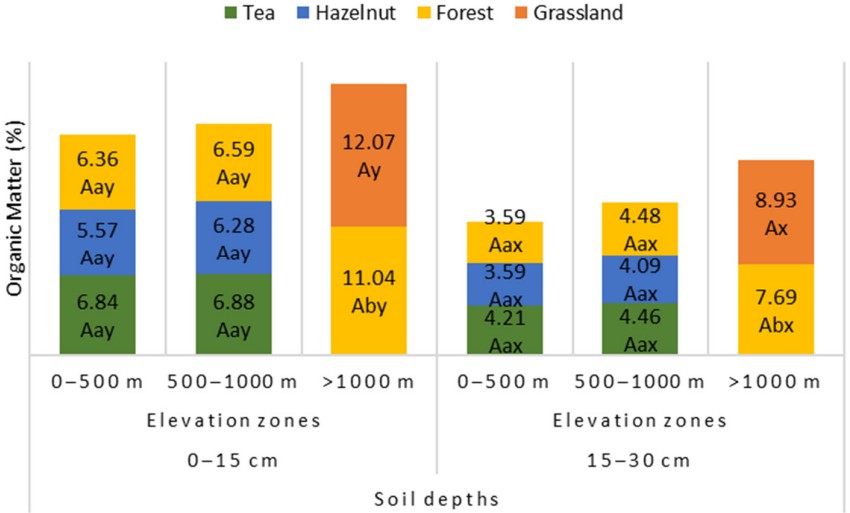

**Figure 8.** Changes in soil organic matter with land use, elevation, and soil depth. Letter A indicates significant differences among the land uses at each depth. Letters a and b indicate variation among the elevation zones at each soil depth. Letters x and y indicate variation among the soil depths in each land use and elevation zone.

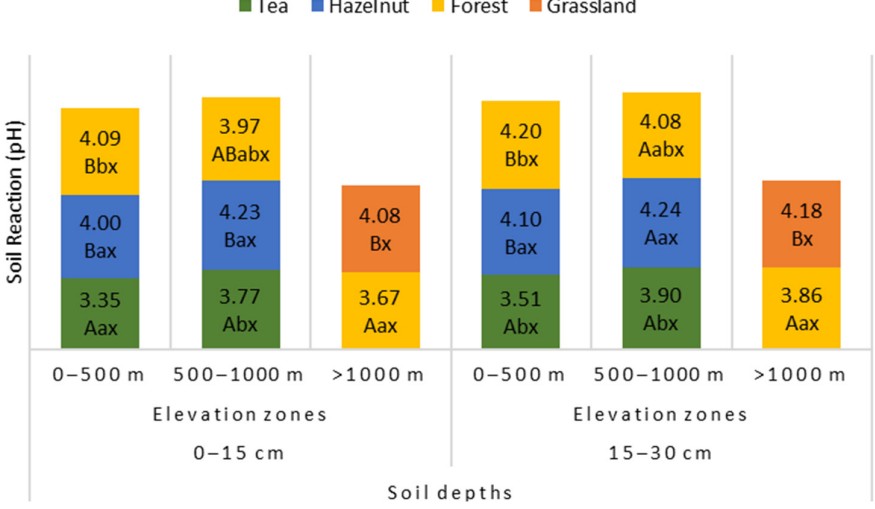

**Figure 9.** Changes in soil pH with land use, elevation, and soil depth. Letters A and B indicate significant differences among the land uses at each depth. Letters a and b indicate variation among the elevation zones at each soil depth. Letter x indicate variation among the soil depths in each land use and elevation zone.

The pH values of soils at 0–15 cm and 15–30 cm depths showed significant differences with elevation zone ($p < 0.05$). The pH values of soils at 0–15 cm soil depths were found to be significantly lower in tea areas located in the 0–500 m elevation zone (3.35) than in

tea areas located in the 500–1000 m elevation zone (3.77) ($p < 0.05$). The pH values of soils at 0–15 cm and 15–30 cm soil depths were found to be significantly lower in forest areas in the >1000 m elevation zone (3.67 and 3.86, respectively) than in forest areas at 0–500 m elevation zone (4.09 and 4.20, respectively) ($p < 0.05$). The results of the correlation analysis revealed significant positive correlations between elevation and pH value in the soils in forest areas ($p < 0.01$) (Table 1). There are many studies indicating that the pH of soils decreases with an increasing elevation, similar to our findings, as a result of increasing rainfall that causes cation leaching in the soil profile [25,27,47].

In general, tea shows optimum growth in soils with pH values between 4.5–6.0 [48]. Accordingly, in this study, the soils in the tea areas were far below the optimum range and had extremely acidic behavior. The fact that the pH value of the tea areas was extremely low compared to other areas might be due to the long-term use of acidity-increasing fertilizers, such as ammonium sulfate, and the acid character of tea residue [49]. When we compare our pH values with those of previous studies, the ratio of the number of samples in which the pH value of Arhavi tea soils was < 4 was 0.12% (2/1725) in 1961 [50], 39.48% (467/1183) in 1983 [51], and 84.85% (56/66) in 2021 (this study). Similarly, Yan et al. [52] and Fan and Han [53] reported that soil pH value in tea areas decreased with tea age.

### 3.5. Electrical Conductivity (µS/cm) (EC)

Variations in the average EC values of soils based on land use, elevation zone, and soil depth levels are given in Figure 10. The EC values of soils at 0–15 cm and 15–30 cm depths showed significant differences with land use ($p < 0.05$). According to Richards' [54] soil EC classification, the soils of the entire study area are in the non-saline electrical conductivity class. Moreover, the EC values of the soils at the 15–30 cm depth level were found to be significantly higher in tea areas (73.6 µS/cm) than in hazelnut (33.5) and forest (43.1) areas in the 0–500 m elevation zone. Kucuk and Yener [5] found similar results in neighboring areas, as well. In a different study performed in Ethiopia, Tellen et al. [40] determined that EC significantly differed with land use.

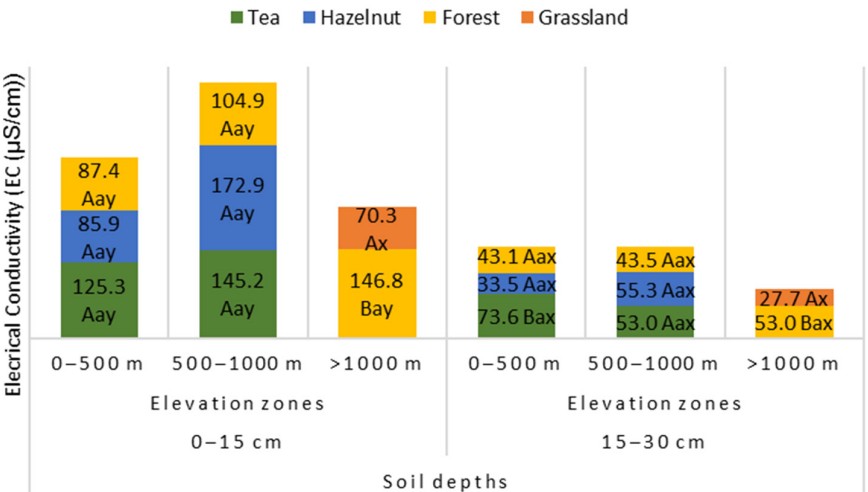

**Figure 10.** Changes in soil electrical conductivity with land use, elevation, and soil depth. Letters A and B indicate significant differences among the land uses at each depth. Letter a indicates variation among the elevation zones at each soil depth. Letters x and y indicate variation among the soil depths in each land use and elevation zone.

The EC values of the soils showed significant differences with elevation zone and soil depth ($p < 0.05$). Pairwise comparisons revealed that the EC values of soils at 0–15 cm and 15–30 cm soil depths were found to be significantly higher in forest areas in the >1000 m elevation zone (146.8 µS/cm and 53.00 µS/cm, respectively) than in forest areas at the 0–500 m (87.4 µS/cm and 43.00 µS/cm, respectively) and 500–1000 m (104.9 µS/cm and 43.5 µS/cm, respectively) zones ($p < 0.05$). The EC values of soils at 0–15 cm depths were

found to be significantly higher than the EC values of soils at 15–30 cm depths in the other land use and elevation zones, except for grassland areas in the >1000 m elevation zone. Tellen et al. [40] determined that EC did not find a significant difference with elevation.

### 3.6. Soil Erodibility Factor (K)

Variation in the average K values of soils with land use, elevation zone, and soil depth levels are given in Figure 11. According to Wischmeier and Smith [55], the soils in the study area were in the class of low erosive soils. The K values of soils did not differ significantly with land use, elevation zone, or soil depth. Similarly, Arunrat et al. [56] and Erol et al. [57] determined that the K factor did not show significant differences with land use. There was found a significant positive correlation between K factor values at 0–15 cm soil depths in grassland areas and elevation ($p < 0.05$) (Table 1). This might be due to a decrease in the amount of organic matter with the increase in altitude in the grassland areas. There was found a significant negative correlation between K values at 0–15 cm and 15–30 cm depths in forest areas and elevation ($p < 0.05$) (Table 1).

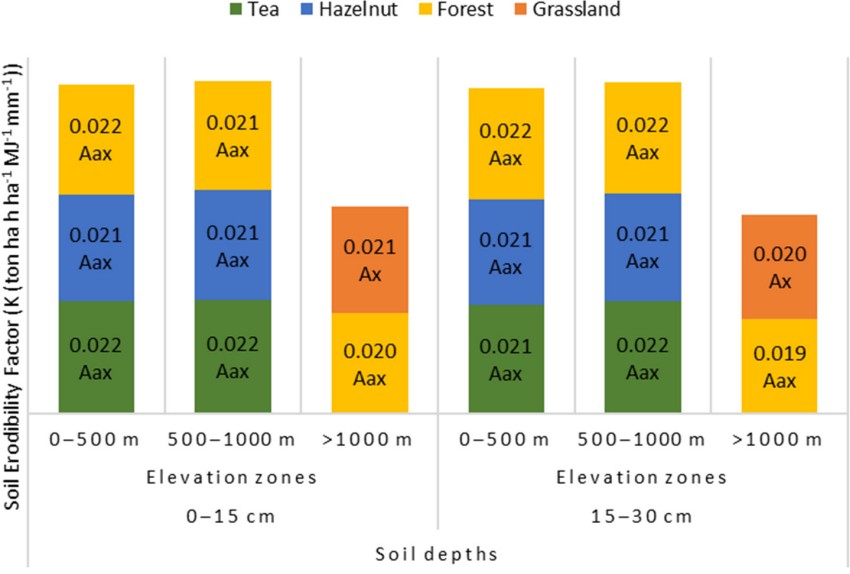

**Figure 11.** Changes in the soil erodibility factor with land use, elevation, and soil depth. Letter A indicates significant differences among the land uses at each depth. Letter a indicates variation among the elevation zones at each soil depth. Letter x indicates variation among the soil depths in each land use and elevation zone.

### 4. Conclusions and Recommendations

The results of this study revealed that the soils in tea and hazelnut gardens, forests, and grasslands in the Arhavi Sub-basin showed significant differences in terms of sand, clay, silt, bulk density, organic matter, and pH properties with land use, soil depth, and elevation. The soils in the study area were in the class of low erosive soils. K values of soils were in the low erosion class and did not differ significantly with land use, elevation zone, or soil depth. An increasing soil acidity was the most important problem in all land uses that was threatening soil fertility. The use of soil acidity-increasing fertilizers, such as ammonium sulfate, in tea gardens should be eliminated in the region to protect the fertility of soils. This study will provide data to future studies related to soil acidification, soil erosion, and land use that are planned in the Arhavi Sub-basin.

**Author Contributions:** A.D.—original draft preparation, methodology, field and laboratory studies, statistical analysis, and writing/reviewing and editing; C.Y.—field and laboratory studies and writing/reviewing and editing; M.T. and A.T.—field studies, writing/reviewing and editing; C.S.—field studies and writing/reviewing and editing. All authors have read and agreed to the published version of the manuscript.

**Funding:** The current work was realized in frames of Joint Operational Black Sea Programme 2014–2020, the project BSB 963 "Protect-Streams-4-Sea", with the financial assistance of the European Union. The content of this publication is the sole responsibility of the authors and in no case should be considered to reflect the views of the European Union.

**Institutional Review Board Statement:** Not applicable.

**Informed Consent Statement:** Not applicable.

**Data Availability Statement:** Not applicable.

**Conflicts of Interest:** The authors declare no conflict of interest. The funders had no role in the design of study; in the collection, analyses, or interpretation of data; in the writing of the manuscript; or in decision to publish the results.

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
