# Peer review of "Variation in Certain Soil Properties Based on Land Use Type, and Elevation in Arhavi Sub-Basin, Artvin, Turkiye"

_sustainability, doi:10.3390/su15119114_

Round 1
Reviewer 1 Report
Authors of the manuscript raised an important and hot topic in determine the changes in certain soil properties with different land uses, soil depth and elevation zones in Turkiye. There is a high need to know more about this issue. However, I have several doubts concerning this manuscript which in details are given at the pdf file.

Although English is not my native language either, it is notable that some verb conjugations and general terms need revision by the authors.
Reviewer 2 Report
At first the article seemed interesting. Perhaps I did not fully understand it, but the results of the research are obvious, the conclusion is short, laconic and does not really bring anything new. I suggest that the authors rethink the concept of this article and perhaps enrich its content.
Reviewer 3 Report
This study examined the changes in soil properties due to different land uses, soil depth, and elevation zones in the Arhavi Sub-Basin, Artvin, Turkiye.
The manuscript is well written. There are still some issues with the document that need to be resolved. Following are some details:
The introduction should be extended, and additional essential research sources should be cited.
On page-2, The latitude and longitude of the study region should be 41 degrees rather than 410.…
On page-3, the Author should include the map Scale in Fig.2
The author should illustrate the methodology framework for this study for a better understanding of the flow of research work.
The author should include the field samples photograph of the different Land use types used in this study.
On Page-4, The author should include a summary of the various methods employed in this study to calculate soil properties.
There is a typo error in many places, the author should correct them.
The author should mention the name of software/tools used for data analysis.
The conclusion section should be expanded by the author to include more logical explanations of the findings and recommendations.
The quality of the English language is good, however, it can be improved for the reader's benefit.
Reviewer 4 Report
Dear Authors
After a detailed reading in the manuscript, titled: "Variation of Certain Soil Properties Based on Land Use Type and Elevation in Arhavi Sub-Basin, Artvin, Turkiye", I recommend ACCEPT the manuscript with minor corrections:
1 - At the end of the Abstract, I could make clear the importance of this study on a global scale.
2 - In Keywords, modify the words that are similar to the title. I suggest the term "great regions" for a broader idea.
3 - At the end of the introduction, it is necessary to report the importance of your study on a global scale, to further arouse the interest of Journal readers.
4 - Figure 1 does not have a good resolution.
5 - The yellow color assigned to Table 1 is not appropriate, just inform in the text, if the numbers are in the table ok, no need to highlight. Remove the yellow from the selection.
6 - The results are adequate. Congratulations to the authors.
7 - It is necessary to expand the conclusion, even with the need to point out possible future studies.
English is well written
Round 2
Reviewer 1 Report
Dear Authors, the effort to meet the demands requested by the me is notable. It appears that recent articles were incorporated into the work in order to support the hypotheses raised by the authors, especially regarding the research gap in the discussion section. The grammar has been extensively modified and improved, meeting the standards required for publication in English.
Regards
Reviewer 2 Report
The revised version of the article is satisfactory.